# A Comprehensive RNA-seq Analysis of Human Bocavirus 1 Transcripts in Infected Human Airway Epithelium

**DOI:** 10.3390/v11010033

**Published:** 2019-01-07

**Authors:** Wei Zou, Min Xiong, Xuefeng Deng, John F. Engelhardt, Ziying Yan, Jianming Qiu

**Affiliations:** 1Department of Microbiology, Molecular Genetics and Immunology, University of Kansas Medical Center, Kansas City, KS 66160, USA; zouw@umich.edu (W.Z.); Xuefeng.Deng@pennmedicine.upenn.edu (X.D.); 2The Children’s Mercy Hospital, School of Medicine, University of Missouri Kansas City, Kansas City, MO 64108, USA; xiongminzw@icloud.com; 3Department of Anatomy and Cell Biology, University of Iowa, Iowa City, IA 52242, USA; john-engelhardt@uiowa.edu (J.F.E.); ziying-yan@uiowa.edu (Z.Y.)

**Keywords:** parvovirus, human bocavirus 1, RNA-seq, transcription profile, human airway epithelia

## Abstract

Human bocavirus 1 (HBoV1) infects well-differentiated (polarized) human airway epithelium (HAE) cultured at an air-liquid interface (ALI). In the present study, we applied next-generation RNA sequencing to investigate the genome-wide transcription profile of HBoV1, including viral mRNA and small RNA transcripts, in HBoV1-infected HAE cells. We identified novel transcription start and termination sites and confirmed the previously identified splicing events. Importantly, an additional proximal polyadenylation site (pA)p2 and a new distal polyadenylation site (pA)d_REH_ lying on the right-hand hairpin (REH) of the HBoV1 genome were identified in processing viral pre-mRNA. Of note, all viral nonstructural proteins-encoding mRNA transcripts use both the proximal polyadenylation sites [(pA)p1 and (pA)p2] and distal polyadenylation sites [(pA)d1 and (pA)d_REH_] for termination. However, capsid proteins-encoding transcripts only use the distal polyadenylation sites. While the (pA)p1 and (pA)p2 sites were utilized at roughly equal efficiency for proximal polyadenylation of HBoV1 mRNA transcripts, the (pA)d1 site was more preferred for distal polyadenylation. Additionally, small RNA-seq analysis confirmed there is only one viral noncoding RNA (BocaSR) transcribed from nt 5199–5340 of the HBoV1 genome. Thus, our study provides a systematic and unbiased transcription profile, including both mRNA and small RNA transcripts, of HBoV1 in HBoV1-infected HAE-ALI cultures.

## 1. Introduction

Human bocavirus 1 (HBoV1) was identified from nasopharyngeal aspirates of pediatric patients with acute respiratory tract infections in 2005 [1,2]. HBoV1 genome has been frequently detected worldwide after respiratory syncytial virus, rhinovirus, and adenovirus infections in hospitalized young children under 2 or 5 years old with acute respiratory tract infections [3,4,5,6,7,8]. Severe and deadly cases, which are associated with high viral loads in respiratory specimens and with anti-HBoV1 IgM antibody detection, increased IgG antibody production, or viremia in serum samples, have been apparently linked to acute respiratory tract infections [9,10,11,12,13,14]. Using RNA-seq to detect the viral RNA genomes and also the viral mRNAs that are transcribed from both RNA and DNA viruses in respiratory secretions, a recent study has identified HBoV1 infections in a significantly greater proportion of patients with community-acquired pneumonia (18.6%) than in controls (2.2%), suggesting that mono-detection of HBoV1 infection is significantly associated with community-acquired pneumonia [15]. In vitro, HBoV1 infects polarized human bronchial airway epithelium (HAE) cultured at an air-liquid interface (ALI; HAE-ALI) and causes damage of the airway epithelium [16,17,18,19,20].

HBoV1 belongs to genus *Bocaparvovirus* of the *Parvoviridae* family. The two prototype members of this genus are minute virus of canines (MVC) and bovine parvovirus type 1 (BPV1) [21,22]. RNA transcription profiles of these three bocaparvoviruses have been well studied through the traditional RNA analysis methods: reverse transcription (RT)-PCR, RNase protection assay (RPA) and Northern blotting [16,21,22,23,24,25,26]. In HBoV1-infected HAE-ALI, through RT-PCR and RPA, we identified novel splicing donor and acceptor sites (D1’ and A1’) of viral pre-mRNA that are used to process HBoV1 mRNAs encoding novel viral nonstructural proteins NS2, NS3, and NS4 [24]. More importantly, we identified the first parvoviral long noncoding RNA (lncRNA), bocavirus-transcribed small RNA (BocaSR), during HBoV1 infection of HAE-ALI, which plays an important role in virus replication [26]. The identification of these novel HBoV1 RNA transcripts highlights the unique features of HBoV1 transcription among parvoviruses. In order to identify additional HBoV1 transcripts and avoid biases imparted by traditional RNA analysis methods, here, we utilized a systemic and unbiased approach to explore the transcription profile of HBoV1 during infection in human airway epithelia. RNA samples extracted from HBoV1-infected HAE-ALI cultures were analyzed with both mRNA-seq and small RNA-seq.

## 2. Materials and Methods

### 2.1. Human Airway Epithelium Cultured at an Air-Liquid Interface (HAE-ALI)

Primary human bronchial airway epithelial cells were isolated from the lungs of healthy donors at the Cell Culture Core of the Center for Gene Therapy, University of Iowa, under a protocol approved by the Institutional Review Board of the University of Iowa (IRB ID No. 9507432). Pseudostratified human airway epithelia as the polarized HAE-ALI cultures were differentiated from the isolated primary cells as previously described [17]. In brief, primary human airway epithelial cells were seeded on collagen-coated, permeable polyester membrane of Costar Transwell^®^ inserts (Cat #3470, Corning, Tewksbury, MA, USA), and then were differentiated (polarized) at an air-liquid interface (ALI) for 3–4 weeks. Ultraser^TM^-G (USG) serum substitute medium (Pall Corporation, Port Washington, NY, USA) was used for the polarization and maintenance of the HAE-ALI. HAE-ALI cultures with a transepithelial electrical resistance (TEER) of >2000 Ω·cm^2^, as determined by Millicell ERS-2 Voltohmmeter (EMD Millipore, Burlington, MA, USA), were used for virus infection in this study.

### 2.2. Virus Infection and Quantification of Apically Released Virions

HBoV1 virions were produced in HEK293 cells transfected with pIHBoV1, a HBoV1 duplex form genome and purified as described previously [17]. HAE-ALI was infected the purified HBoV1 stock at a multiplicity of infection (MOI) of 1000 viral genome copy number (vgc)/cell, as described previously [17]. At 7 days post-infection, an aliquot of 100 μL of phosphate buffered saline (PBS), pH7.4, was added to the apical chamber of the HAE-ALI culture, and harvested as an apical wash. The presence of progeny virions in the apical washes was an indicator of the productive HBoV1 infection in HAE-ALI. All the apical washes were stored at 4 °C for quantification of vgc using a quantitative PCR (qPCR) with HBoV1-specific primers and probe, following the method described previously [17].

### 2.3. Immunofluorescence Analysis

We fixed a small piece of the insert membrane with 3.7% paraformaldehyde in PBS, and permeabilized with 0.2% Triton X-100 in PBS, followed by direct staining with an anti-HBoV1 NS1C antibody [23]. Immunofluorescence analysis was performed using a method described previously [17]. Confocal images were taken with an Eclipse C1 Plus confocal microscope (Nikon, Tokyo, Japan) controlled by Nikon EZ-C1 software. DAPI (4′,6-diamidino-2-phenylindole) was used to stain the nucleus.

### 2.4. RNA Extraction

At 7 days post-infection, mock- and HBoV1-infected cells were collected for total RNA isolation using miRNeasy Mini Kit (Qiagen, Valencia, CA, USA) following the manufacturer’s instructions with DNase I treatment. HAE cells from two Transwell^®^ inserts were used to prepare one RNA sample. A total of 6 RNA samples from HBoV1-infected HAE-ALI cultures (HBoV1 group) and 6 RNA samples from mock-infected HAE-ALI cultures (Mock group) were prepared. The quality of RNA samples was analyzed on an Agilent 2100 Bioanalyzer using an Agilent RNA 6000 Nano Kit for an RNA Integrity Number (RIN). Three RNA samples with RINs ≥ 8.0 were chosen randomly from each group and used for mRNA-seq and small RNA-seq.

### 2.5. mRNA-seq and Small RNA-seq

mRNA-seq was performed at Otogenetics Corporation (Atlanta, GA, USA). For mRNA-seq, the TruSeq Stranded Total RNA with Ribo-Zero HMR library Prep Kit (#RS-122-2201, Illumina, San Diego, CA, USA) was used to prepare the sequencing library from 1 μg of total RNA, and 2 × 106 bp paired-end sequencing in high output run mode was performed using Illumina HiSeq 2500 system. Small RNA-seq was performed at SeqMatic Company (Fremont, CA, USA). TailorMix miRNA V2 library preparation kit was used to prepare the small RNA library. 2 × 150 bp paired-end sequencing was performed using Illumina MiSeq system (San Diego, CA, USA).

### 2.6. mRNA-seq Read Mapping and Junction Analysis

The resulting base calling (bcl) files of mRNA-seq were converted to FASTQ files using Illumina’s bcl2fastq v2.17.1.14 software (San Diego, CA, USA). TopHat 2.0.9 was used to map mRNA-seq reads against the HBoV1 full-length genome (GenBank accession no: JQ923422) using default parameters. Samtools were used to convert a BAM (.bam) file to a SAM (.sam) file. The HBoV1 mapping reads were extracted using an in-house-developed script. The alignment data were used to build the junction tracks by Integrative Genome View (IGV; http://software.broadinstitute.org/software/igv/home). The junctional events were identified only when a single read splits across two exons.

### 2.7. Detection of Polyadenylation Sites

We searched the paired-end mRNA-seq reads for polyadenine repeats of a length of 9 (i.e., AAAAAAAAA for R2, and TTTTTTTTT for R1), and then mapped those reads with the polyadenine repeats to the HBoV1 full-length genome by BWA (Burrows-Wheeler Aligner). It is noted that HBoV1 genome contains no polyadenine repeats longer than 8.

### 2.8. HBoV1 RNA Transcripts Assembling

Three samples FASTQ files were merged together. TopHat 2.0.9 was used to map RNA-seq reads against the HBoV1 full-length genome using default parameters. Based on the previous identified transcripts, splicing junctions and polyadenylation signals in the present study, we listed all possible HBoV1 mRNA transcripts as the reference viral transcript profile. Transcript assembly and abundance estimation were conducted with StringTie software and reported in Fragments Per Kilobase of exon per Million fragments mapped (FPKM).

### 2.9. Mapping of Small RNA-seq Reads and Transcript Abundance Estimation

The CAP-miRSeq v1.123 pipeline was employed for read pre-processing, alignment, mature/precursor/novel miRNA detection and quantification. In this pipeline, Cutadapt was used to trim reads adaptor at the 3′-end. After adaptor trimming, reads length less than 17 bp were discarded. Then the reads were mapped to the full-length HBoV1 genome by mapper, identified as known and novel miRNA by miRDeep2, and quantified expression by quantifier. Meanwhile, we used Bowtie 2 to align small RNA-seq reads to HBoV1 genome for identification of all small RNA transcripts. The count of HBoV1 mapping reads was extracted using an in-house-developed script from an alignment file. The alignment data were used to build the coverage tracks by Integrated Genome Browser (IGB; http://igb.bioviz.org/).

## 3. Results

### 3.1. Virus Infection

To ensure a high infectivity, HAE-ALI cultures were infected with HBoV1 at an MOI of 1000 vgc/cell. At 7 days post-infection, immunofluorescence analysis revealed that majority HAE cells were infected (Figure 1), and qPCR for HBoV1 genome in the apical washes found that HBoV1 virions released from the apical side of the HAE-ALI culture reached a level of >1.5 × 10^11^ gc/mL, representing effective HBoV1 infections in HAE-ALI cultures [18]. Total RNA samples for next-generation RNA sequencing were extracted from these highly infected HAE-ALI cultures.

### 3.2. Illumina mRNA-seq Next-Generation Sequencing

We chose three total RNA samples isolated from HBoV1- or mock-infected HAE cells that had RIN values of ≥8 for Illumina mRNA-Seq. The complete raw and normalized mRNA-seq data have been deposited in the Gene Expression Omnibus of the National Center for Biotechnology Information (accession no. GSE102392). Reads between 71.8 and 92.3 million were obtained by Illumina RNA-Seq (Table 1). Approximate 75% of total reads were successfully mapped either to the human genome or to the HBoV1 genome. In three repeated RNA samples extracted from mock-infected HAE-ALI, all reads were mapped to the human genome except less than 20 reads that were mapped to the HBoV1 genome. In three repeated HBoV1 RNA samples, 0.21–0.47 million reads (0.23–0.55%) were mapped to the plus strand of the HBoV1 genome. Of note, there were also a few reads (9458–21,580) mapped to the minus strand of the viral genome, indicating the potential transcription capability of the minus strand. However, in this study, we only focused on these reads from the positive strand.

### 3.3. mRNA-seq Reads Mapping on the HBoV1 Plus Strand

We analyzed the reads mapped to the plus strand of HBoV1 genome. By sequence alignments, a coverage map of the HBoV1 genome was created, which displays the number of reads that are mapped to a specific position of the HBoV1 genome (Figure 2A). Although there were some variations between the three repeats in terms of total reads, we obtained a similar trend in the read coverages across the biologic replicates. Steep increases of mRNA-seq read counts are indicative of either transcription initiations or splicing events.

From the mRNA-seq reads mapped to the HBoV1 full-length genome, it showed that viral mRNA transcripts initiated as early as nt 80, but at an abundance much lower than reads initiating at nt 291–296 (Figure 2B). These results confirmed transcriptional initiation from P5 promoter at nt 291–296, which was close to the previously determined initiation site at nt 282 [23], and suggest that the left-end hairpin (LEH) contains properties of promoter activity; this property is similar to what has been observed with the inverted terminal repeats (ITRs) of adeno-associated viruses (AAVs) [27,28]. On the right-hand hairpin (REH) end, all the three HBoV1 RNA samples contained viral mRNAs that ended as far as to nt 5499 (Figure 2C), suggesting an alternative distal polyadenylation signal. The results also confirmed the major mRNA transcripts ended at the previously identified polyadenylation site, (pA)d1, at nt 5171 immediately after capsid proteins (VP)-coding sequence [23].

### 3.4. Analyses of Alternative Splicing of the HBoV1 pre-mRNA

Previously, we and other groups had shown that there were six introns in the single HBoV1 pre-mRNA [16,23,24]. mRNA-seq reads confirmed all the six splice junctions including D1/A1’, D1’/A1, D1/A1, D1/A2, D2/A2, and D3/A3 (Figure 3 and Table 2). The junction reads revealed that the second intron splicing (D2/A2) had the highest frequency, which is consistent with a previous study showing that nearly 75% HBoV1 mRNAs splice out the second intron [23]. There was also a relatively high frequency of junction reads at the D3/A3 sites, splicing of which is required for production of VP-expressing mRNAs. However, the reads at the junctions for D1 to A1’ and D1’ to A1 splicing events were relatively low, indicating the virus only expresses few mRNAs that encode NS2, NS3, and NS4 [24]. Of note, two novel splicing events between sites of nt 337 and nt 1108 and sites of nt 337 and nt 2198 were identified in all three RNA samples of HBoV1 infection. The two novel acceptor sites nt 1108 and nt 2198 are close to the A1’ (nt 1017) and A1 (nt 2140) acceptor sites, respectively. There were also other novel splicing events that were detected in two or one of the samples (Table 2). Although all newly identified splicing events could produce novel HBoV1 transcripts, these spliced mRNAs have unchanged open reading frames (ORFs). Of note, we did not find any splicing events from D1 to A3 sites, which we previously predicted to produce VP-expressing mRNA R8 [23,25], indicating that R6 and R7 VP-expressing mRNAs are the only mRNA transcripts for production of capsid proteins.

### 3.5. Analyses of Alternative Polyadenylation in the HBoV1 pre-mRNA

HBoV1 pre-mRNA is polyadenylated at (pA)p and distal (pA)d polyadenylation sites, although the mechanism controlling polyadenylation choice remains unclear [23]. There are five consensus polyA signals (CPSF160-binding site AAUAAA) [29] that can be used for polyadenylation at the proximal site, which are located at nt 3295–3300, nt 3329–3334, nt 3409–3414, nt 3440–3445, and nt 3485–3490 [25]. There is only one AAUAAA site located at the end of the genome (nt 5153–5158). By searching polyA sequences [>(A)9] in the mRNA-seq reads and mapping these reads containing a sequence of nine or more A residues to HBoV1 genome, we found that AAUAAA at nt 3329 and the previously identified AAUAAA at nt 3485 were most commonly used for proximal polyadenylations at the (pA)p2 and (pA)p1 sites, respectively (Table 3). While the previous reported distal polyadenylation site (pA)d1 at nt 5171 [23] was confirmed, our analyses also found a novel (pA)d_REH_ at the REH region where HBoV1 mRNA transcripts were cleaved and polyadenylated (Table 3). Of note, instead of adding polyA at a specific nucleotide, the polyadenylation at (pA)d_REH_ occurs in a wide range of 130 nts from nt 5369 to 5499 with hot sites at nt 5443 and 5444 (Figure 3).

### 3.6. Summarized HBoV1 mRNA Transcripts by RNA-seq

Due to the fact that different HBoV1 mRNA transcripts share the same exon sequences, but the alternative usage of the introns and alternative polyadenylation sites, it is hard to assemble de novo HBoV1 mRNA transcripts, based on the mRNA-seq reads. Thus, we used the newly identified (pA)p2 and (pA)d_REH_ and the previously identified mRNA transcripts as templates to assemble HBoV1 mRNA transcripts using the mRNA-seq data. The results showed that almost all NS- and NP1-coding mRNA transcripts used all four polyadenylation sites (pA)p1, (pA)p2, (pA)d1, and (pA)d_REH_; whereas VP-expressing mRNAs only used (pA)d1 and (pA)d_REH_ sites (Figure 4). All NS- and NP1-coding mRNAs showed the same expression levels at the (pA)p sites, (pA)p1 vs (pA)p2, as well as the (pA)d sites, (pA)d1 vs (pA)d_REH_. Of note, they utilize the (pA)p site [(pA)p2 + (pA)p1] 1.5–2.5-fold more frequently than the (pA)d sites [(pA)d1 + (pA)d_REH_] (Table 3 and Figure 4).

NS1-coding transcripts spliced at D2/A2 sites were expressed at the highest level followed by NP1-, NS2- and NS4-coding transcripts in order, while NS1-70 and NS3 mRNAs were expressed at lower levels (Table 3). NS2 transcripts contain mRNA spliced at D1’/A1 and D2/A2 sites and mRNA spliced at D1’/A1 site, and the former mRNA showed an expression level twice more than the second one. NS3 transcripts contained two splicing forms, mRNA spliced at D1/A1’ and D2/A2 sites and mRNA spliced at D1/A1’ sites. NS4 transcripts contained mRNA spliced at D1/A1’, D1’/A1, and D2/A2 sites, and mRNA spliced at D1/A1’ and D1’/A1 sites. NP1 transcripts also had two splicing forms, mRNA spliced at D1/A1 and D2/A2 sites and mRNA spliced at D1/A2 site. The two splicing forms of NS3, NS4 and NP1 transcripts showed the similar expression level. All these NS- and NP1-coding mRNAs were polyadenylated at (pA)d1 and (pA)d_REH_ sites at a similar ratio (Table 3 and Figure 4). VP-coding mRNA transcripts contain two spliced forms, mRNA spliced at D1/A1, D2/A2, and D3/A3 sites, and mRNA spliced at D1/A2 and D3/A3 sites. Importantly, the VP-coding mRNA transcripts polyadenylated at the (pA)d1 site was at a level of approximately 7-fold higher than those polyadenylated at (pA)d_REH_ (Table 3 and Figure 4).

### 3.7. Small RNA-seq Analysis

The complete raw and identified viral small RNA read counts of small RNA-seq data have been deposited in the Gene Expression Omnibus of the National Center for Biotechnology Information (accession no. GSE GSE123253). Data of the small RNA-seq does not suggest that HBoV1 produce any classical miRNA, however, it revealed a small RNA hot spot located between nt 5199–5340 (Figure 5), which confirmed the expression of BocaSR encoded at nt 5199–5338 [26]. Of note, in addition to the full-length small RNA between 5199–5340, we also identified many smaller RNAs with different lengths located within this region (Table 4). However, we did not detect any of these smaller RNAs by Northern blotting using high percentage of polyacrylamide gels (data not shown), suggesting that these smaller RNAs detected by small RNA-seq might be artifacts of sequencing due to the secondary structure. Taken together, our results confirm that HBoV1 express small RNAs from the 3’ end of the genome at nt 5199–5340, but not from anywhere else of the viral genome.

## 4. Discussion

Next generation RNA sequencing technology has been widely used in many aspects of biological research, including virology for the study of gene expression by both the host and virus. This technology is a powerful tool for the systematic and unbiased profiling of viral transcription/expression. Both mRNA-seq and small RNA-seq have been applied to profile gene expression of AAV2 in the presence or absence of coinfection of helper viruses, which identified novel AAV2 mRNA transcripts from both the positive and negative stands of the AAV2 genome, as well as viral miRNAs [30]. In the present study, we utilized mRNA-seq and small RNA-seq of RNA samples isolated from HBoV1-infected HAE-ALI cultures to study the transcription profile of HBoV1. Through this study, we have mapped HBoV1 transcriptional initiation sites, usage of different splicing donor and acceptor sites, and alternative polyadenylation sites. Importantly, the mRNA-seq also provided quantitative information on various HBoV1 mRNA transcripts, which will be useful for understanding HBoV1 transcription regulation from a single promoter.

**Transcription start site.** We previously identified the HBoV1 P5 promoter starts transcription at nt 282 from total RNA extracted from termini-less HBoV1 plasmid-transfected HEK293 cells [23], which was confirmed by this mRNA-seq. Additionally, our data also showed that a portion of HBoV1 mRNAs are transcribed as early as nt 80, which is located in the LEH region, suggesting the presence of cryptic promoter activity in the LEH. Interestingly, the LEH region of bocavirus BPV1 and the AAV ITRs also have promoter activities [22,27,28]. The reads covering nt 80–282 demonstrated a Poisson distribution (Figure 2). These extended 5’ untranslated regions (UTR) may also contain regulatory functions, though the abundance of these early mRNA transcripts is quite low.

**Alternative RNA splicing sites.** Pre-mRNA splicing is the key strategy that HBoV1 applies to generate different mRNA transcripts since all HBoV1 transcripts are generated from one pre-mRNA. There are nearly twenty matured mRNAs generated to encode nonstructural proteins (NS1/NS1-70, NS2/NS2’, NS3/NS3’, NS4/NS4’, and NP1) and structural proteins (VP1, VP2, and VP3) (Figure 4). The expression levels of NS2’, NS3’, and NS4’ were too low to be detected by Western-blotting [24]. Two novel donor sites (nt 1201 and nt 2423) and five acceptor sites (nt 1108, nt 1118, nt 2187, nt 2198 and nt 2576) were identified in at least two repeated RNA samples. The novel donor site at nt 1201 is closed to D1’ (nt 1212), and the one at nt 2423 is closed to D3 (nt 2453). Of note, both novel donor sites do not contain the canonical donor sequence of GU. All five novel acceptor sites contain classical AG sequence except the one at nt 2187 site. The acceptors at nt 1108 and nt 1118 are close to the A1’ site (nt 1017), and the ones at nt 2187 and nt 2198 are close to the A1 acceptor at nt 2140. Interestingly, the D1 donor site used three novel acceptor sites at nt 1108, nt 1118 and nt 2198. There are also mRNA transcripts spliced at D1’ donor and the acceptor at nt 2198. The donor at nt 1201 used the acceptor at nt 2187, and the donor at nt 2423 used the acceptor at nt 2576. Nevertheless, all these mRNA transcripts processed at these novel splice sites are predicated not to produce novel proteins as they do not have novel ORFs. We speculate that they may serve as backup mRNA transcripts that will be used in case the original donor or acceptor sites are mutated during virus replication. Therefore, these alternative donor and acceptor sites can still generate mRNA transcripts for production of viral proteins in the events that the viral genome has mutations.

**Alternative polyadenylation**. Both the NP1 of HBoV1 and MVC regulate proximal polyadenylation to facilitate production of VP-coding mRNAs, which ensures a productive infection [25,31,32,33]. We identified an additional (pA)p site [(pA)p2] used for proximal polyadenylation, in addition to the previously identified proximal polyadenylation site [called (pA)p1 here] [23], highlighting the complexity of internal polyadenylation regulation by the NP1 protein. The mRNA-seq data showed a ratio of 1.08 that the mRNAs are polyadenylated at the sites (pA)p2 vs (pA)p1, suggesting that HBoV1 mRNA is polyadenylated at the (pA)p2 site with a roughly equal efficiency at the (pA)p1 site, consistent with a previous report [34].

In addition to the previously identified (pA)d site at nt 5171 [(pA)d1], we identified a stretch of distal polyadenylation sites [(pA)d_REH_] ranging from nt 5369 to nt 5499 with peak sites at nt 5443 and nt 5444, which was located in the REH. The (pA)d_REH_ might be a universal feature of bocaviral mRNA polyadenylation. In fact, bocaparvoviruses MVC and BPV1 utilize only the distal polyadenylation sites located at similar positions on the REH (close to the turn-round or the loop) [21,22]. However, there are no AAUAAA or degenerative polyA signals presented on or close to the HBoV1 REH. We hypothesize that the bocavirus REHs intrinsically contain strong upstream and downstream polyadenylation signals [35], and HBoV1 (pA)d_REH_ contains a non-canonical polyadenylation signal that helps addition of polyA resides at a long stretch of cleaved RNA ends that read through the (pA)d1. The feature that HBoV1 mRNAs utilize two polyadenylation sites at the distal end is unique among all parvoviruses, which warrants further characterization.

It has been reported that HBoV1 mRNAs, which were generated from transfection of a full-length HBoV1 clone in HEK293 cells, were polyadenylated dominantly at the (pA)d_REH_ site at nt 5445 [36]. However, analyses of the mRNA-seq on the RNA samples derived from HBoV1 infection in HAE-ALI suggests that the majority of HBoV1 mRNAs are distally polyadenylated at (pA)d1, with a ratio of the total RNA reads at (pA)d1 vs. (pA)d_REH_ of 2.09. We speculate that viral mRNA transcripts generated from a transfected plasmid are polyadenylated differently from those produced during virus infection.

**HBoV1 mRNA transcripts**. Identification of novel (pA)p2 and (pA)d_REH_ polyadenylation sites make the HBoV1 mRNA transcripts more diversified. Our previous study identified that NP1 plays an important role in regulating splicing at the A3 acceptor site and facilitates HBoV1 mRNAs to read through the (pA)p sites for production of VP-coding mRNAs [25]. We found that NP1 transcripts used (pA)p1 vs. (pA)p2 and (pA)d1 vs. (pA)d_REH_ equivalently. However, NS1-4 transcripts are preferred to use the proximal polyadenylation sites than the distal polyadenylation sites.

Although the mRNA-seq showed that NS1 transcripts are expressed at the highest level among these NS1-4-coding transcripts, the NS1 protein is expressed at a much lower level than NS2-4 in HBoV1-infected HAE cells [24]. The details of this regulatory mechanism warrant further investigation. It is hard to determine the protein expression levels of NS2 and NS3 in HBoV1- infected HAE cells because of their close molecular weight [24]. The fact that NS2 transcripts are expressed at a much higher level than NS3 transcripts may be associated with the observation that NS2 is indispensable to virus replication in HAE cells, whereas NS3 is dispensable [24].

**HBoV1 small RNA transcripts**. Recently, we identified a HBoV1 long noncoding RNA (BocaSR) located between nt 5199–5338 [26]. BocaSR regulates the expression of HBoV1-encoded non-structural proteins NS1, NS2, NS3, and NP1, but not NS4. BocaSR accumulates in the viral DNA replication centers within the nucleus and also plays a direct role in replication of the viral DNA [26]. The small RNA-seq further confirmed the expression of BocaSR. Interestingly, except for the BocaSR, we identified many smaller RNAs spanning this region; however, we did not find any canonical secondary structure in these detected smaller RNAs that might be suggestive of biologic function. Currently, we are unsure whether these detected small RNAs are degraded RNAs of BocaSR or artifacts of small RNA-seq.

In conclusion, we used both mRNA-seq and small RNA-seq data to profile RNA transcripts during HBoV1 infection of human airway epithelia. We identified novel polyadenylation sites not previously observed in other studies [34,36]. By using the previously established transcription map of HBoV1 [2], we were able to assemble a *de novo* HBoV1 transcription map. Small RNA seq data suggest that there is likely one viral small RNA transcribed during infection. Thus, our study provides an example to determine a comprehensive and unbiased parvoviral transcription profile using RNA-seq data.

## Figures and Tables

**Figure 1 viruses-11-00033-f001:**
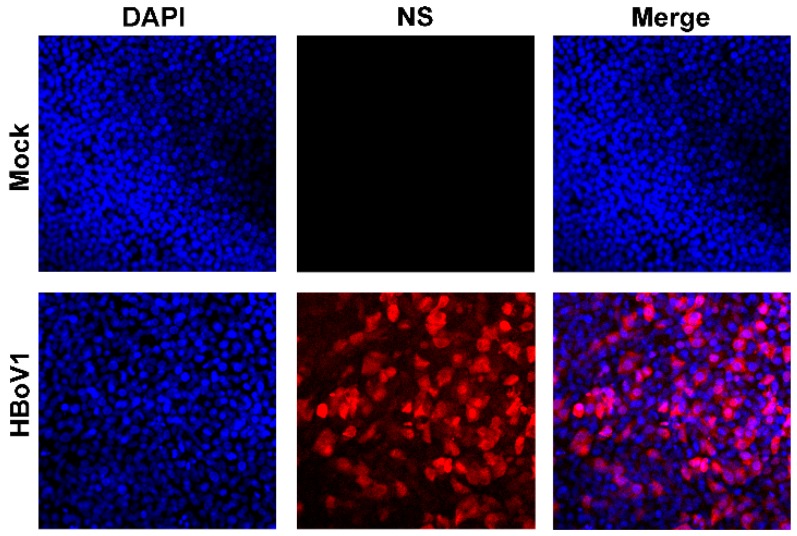
HBoV1 infection of HAE-ALI cells. HAE-ALI cultures were infected with HBoV1 at an MOI of 1000 vgc/cell, or mock-infected. At 7 days post-infection, the pieces of the inserts were fixed and subjected to direct immunofluorescence analysis. The membranes were stained with anti-HBoV1 NS1C antibody that detected all NS1-4 proteins. Images were taken using an Eclipse C1 Plus (Nikon) confocal microscope under 40×, which was controlled by Nikon EZ-C1 software. The nuclei were stained with DAPI (blue).

**Figure 2 viruses-11-00033-f002:**
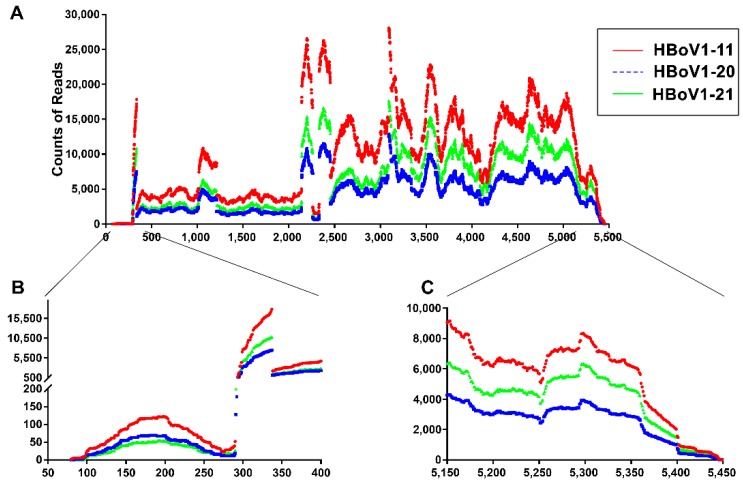
Coverage plots of viral mRNA-seq reads mapped to the HBoV1 genome. Red, blue and green lines represent three RNA samples of HBoV1-infected HAE-ALI cultures. (**A**) Reads mapped to HBoV1 full-length genome. The reads mapped to HBoV1 full-length genome are shown with the coverage across the entire HBoV1 genome (GenBank accession no: JQ923422). (**B** and **C**) Read coverages of the left- and right-hand ends. The read coverages of the left (nt 50–400) (**B**) and right (nt 5150–5450) (**C**) ends of the viral genome are enlarged for details.

**Figure 3 viruses-11-00033-f003:**
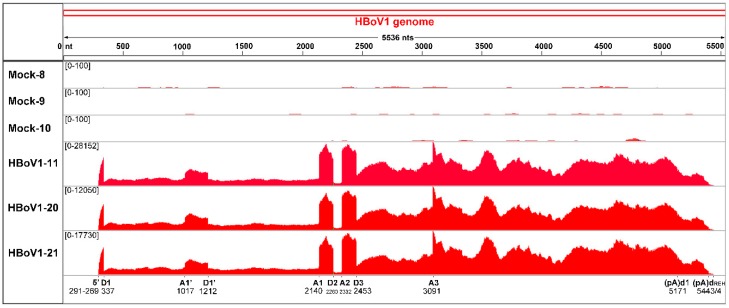
Analysis of alternative splicing of HBoV1 pre-mRNA. The mRNA-seq reads of three HBoV1-infected (HBoV1-11, 20, and 21) and three mock-infected (Mock-8, 9, and 10) RNA samples are mapped to HBoV1 genome. Histograms show the frequency of mRNA reads across the coverage of the HBoV1 genome. The junction events were identified only when at least a single read splits across two exons. The identified splicing junctions are labeled at the bottom with nucleotide numbers shown. The range of reads numbers are shown in each sample. The reads number of all identified splicing events including novel identified splicing and the donor and acceptor sites are shown in Table 2.

**Figure 4 viruses-11-00033-f004:**
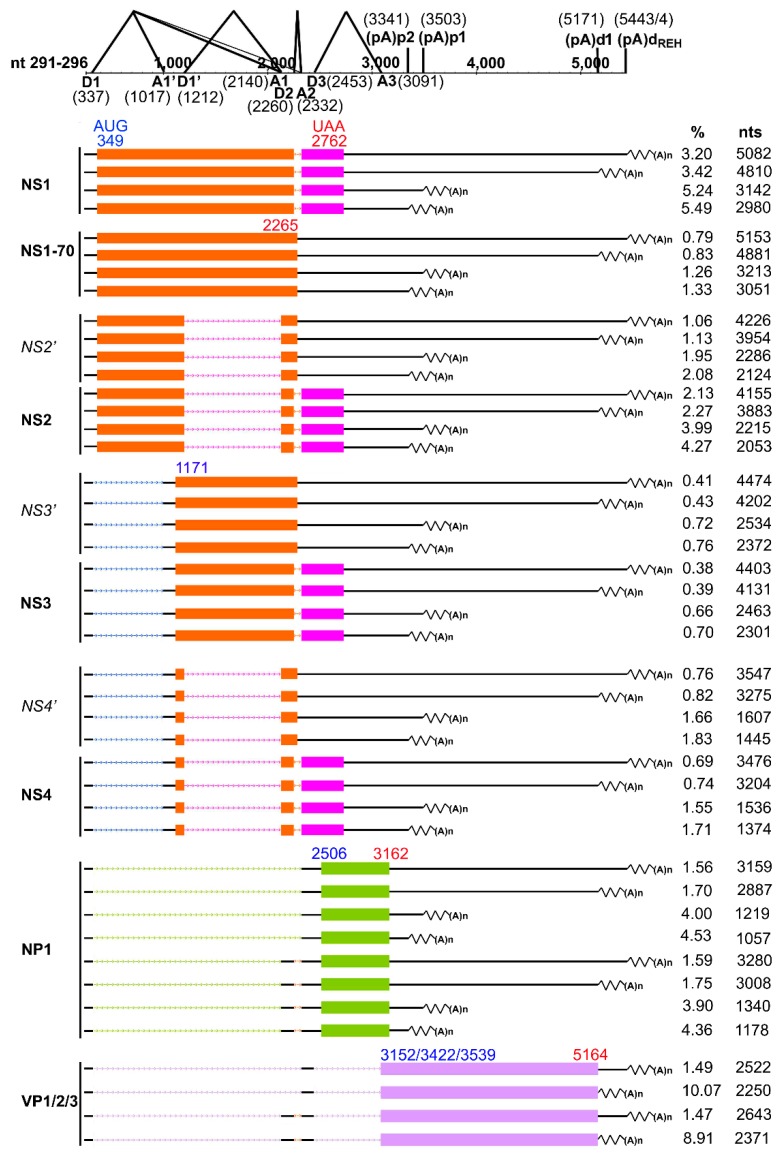
Analyses of all HBoV1 mRNA transcripts based on mRNA-seq data. HBoV1 genome is schematically diagramed with identified donor and acceptor sites, as well as the identified (pA)p1, (pA)p2, (pA)d1 and (pA)d_REH_ polyadenylation sites, used during virus infection. Assembled transcripts of NS1, NS1-70, NS2, NS3, NS4, NP1, and VP using alternative splicing sites and alternative polyadenylation sites [(pA)p1, (pA)p2, (pA)d1 and (pA)d_REH_] are diagramed. Percentage (%) of each transcript in total and the length of each transcript (nt), as determined from the initiation site at nt 291 to the cleavage site (minus the polyA tail), are shown to the right side. Boxes in color indicate ORFs of the exons. Nucleotide numbers of start (AUG; GUG at nt 3422 for VP2) and stop (e.g., UAA) are shown in the first appeared ORF. Introns in the mRNA transcript are diagrammed with dot lines.

**Figure 5 viruses-11-00033-f005:**
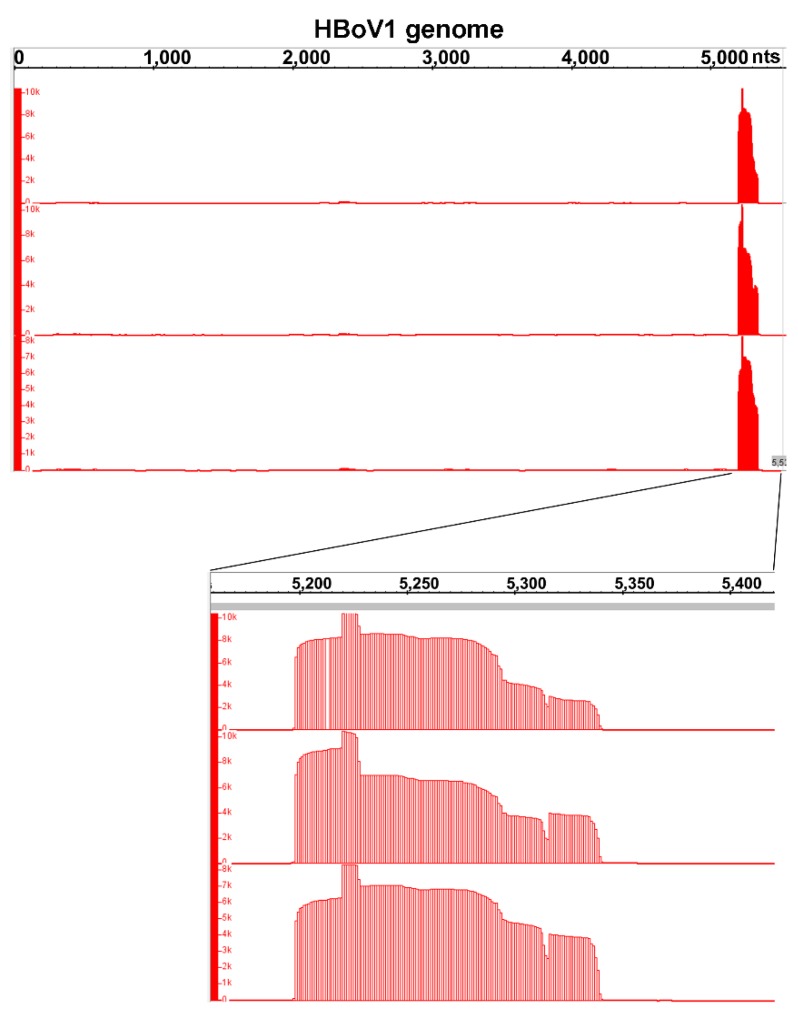
Coverage plots of viral small RNA-seq reads mapped to the HBoV1 genome. The small RNA-seq reads of three HBoV1-infected RNA samples are mapped to HBoV1 genome. Histograms show the frequency of small RNA reads that have the coverage across the HBoV1 genome. The reads coverage of the right ends (nt 5150–5400) of the HBoV1 genome are enlarged for detailed information, show as histograms at the bottom.

**Table 1 viruses-11-00033-t001:** Summary of mRNA-seq results.

Groups	Total Reads	No. (%) of Mapped Reads	No. (%) of Human Reads	No. (%) of HBoV1 reads		
				Plus Strand	Minus Strand	Ambiguous
Mock-8	92,255,534	69,845,826 (75.71)	69,845,805 (75.71)	19 (0.00)	2 (0.00)	0 (0.00)
Mock-9	83,459,160	63,548,966 (76.14)	63,548,950 (76.14)	15 (0.00)	1 (0.00)	0 (0.00)
Mock-10	95,389,110	72,602,804 (76.11)	72,602,787 (76.11)	17 (0.00)	0 (0.00)	0 (0.00)
HBoV1-11	86,814,188	64,362,968 (74.14)	63,865,552 (73.57)	474,990 (0.55)	21,580 (0.02)	846 (0.00)
HBoV1-20	90,567,600	68,650,250 (75.80)	68,429,393 (75.56)	208,998 (0.23)	9458 (0.01)	56 (0.00)
HBoV1-21	7,175,830	54,487,120 (75.93)	54,162,045 (75.48)	309,318 (0.43)	12,952 (0.02)	142 (0.00)

**Table 2 viruses-11-00033-t002:** Detected splicing events in HBoV1 infected HAE cells.

Donor	Acceptor	Splicing Events	Strand	HBoV1-11	HBoV1-20	HBoV1-21	Ratio ^#^ (%)
337 ^§^	1016	D1/A1’	+	2349	1081	1334	85.10
337	1060		+	35	0	0	
337	1108		+	327	144	160	14.90
337	1118		+	101	0	67	
337	2139	D1/A1	+	7962	2985	4698	97.82
337	2198		+	158	83	107	2.18
337	2331	D1/A2	+	4734	2084	3124	99.4
337	861		+	30	0	0	0.60
337	998		+	0	0	30	
1212	2139	D1’/A1	+	3818	1462	1814	95.12
1212	2198		+	0	89	111	4.88
1201	2187		+	0	52	112	
1212	2331	D1’/A2	+	95	71	46	14.73
1212	2372		+	414	0	0	
1630	2139		+	644	0	0	85.27
2250	2362		+	169	0	0	
2260	2331	D2/A2	+	14,175	6222	8782	98.86
2423	2576		+	228	107	0	1.14
2453	3090	D3/A3	+	12,973	5794	9006	99.44
3180	3865		+	0	0	156	0.56

^§^ Nucleotide numbers of the HBoV1 genome. ^#^ The ratio was only counted based on single splicing events.

**Table 3 viruses-11-00033-t003:** Assembled HBoV1 mRNA transcripts and expression level.

Transcripts (Splicing form)	mRNA Polyadenylation Sites			
	(pA)p1	(pA)p2	(pA)d1	(pA)d_REH_
NS1 (D2A2)	15,342	16,071	10,029	9364
NS70	3686	3883	2440	2303
NS2 (D1’A1/D2A2)	11,700	12,500	6646	6243
NS2 (D1’A1)	5721	6105	3313	3098
NS3 (D1A1’/D2A2)	1938	2061	1155	1101
NS3 (D1A1’)	2102	2227	1268	1190
NS4 (D1A1’/D1’A1/D2A2)	4549	4998	2181	2007
NS4 (D1A1’/D1’A1)	4870	5350	2397	2213
NP1 (D1A1/D2A2)	11,434	12,782	5114	4667
NP1 (D1A2)	11,724	13,269	4971	4580
VP (D1A1/D2A2/D3A3)	NA	NA	26,107	4306
VP (D1A2/D3A3)	NA	NA	29,504	4378

Note: NA, not available.

**Table 4 viruses-11-00033-t004:** Top 20 identified small RNAs by small RNA-seq.

Start	End	Length	Strand	Sample 1 (Reads)	Sample 2 (Reads)	Sample 3 (Reads)	Total (Reads)
5199	5228	29	+	373	664	277	1314
5317	5340	23	+	316	532	421	1269
5221	5315	94	+	356	284	306	946
5221	5314	93	+	198	326	362	886
5199	5295	96	+	415	235	179	829
5199	5229	30	+	253	395	165	813
5199	5293	94	+	386	202	158	746
5199	5221	22	+	128	297	216	641
5200	5228	28	+	131	184	85	400
5317	5336	19	+	97	151	129	377
5317	5339	22	+	90	168	108	366
5199 ^#^	5339	140	+	37	94	148	279
5221	5317	96	+	135	61	80	276
5199	5294	95	+	109	88	71	268
5317	5341	24	+	63	122	82	267
5199	5290	91	+	111	72	61	244
5199	5227	28	+	76	120	40	236
5221	5313	92	+	45	69	118	232
5199	5338	139	+	21	67	134	222
5199	5340	141	+	34	72	113	219

^#^ Numbers in red are indicative of coverages of the entire BocaSR.

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
