# Peer review of "A Comprehensive RNA-seq Analysis of Human Bocavirus 1 Transcripts in Infected Human Airway Epithelium"

_viruses, 2019, doi:10.3390/v11010033_

Round 1
Reviewer 1 Report
The current manuscript entitled: “A comprehensive RNA-seq analysis of human
bocavirus 1 transcripts in infected human airway
epithelium.” tackles the complex transcription pattern of human bocavirus a member of the family Parvoviridae. This is an interesting study using top-of-the-line technology in order to fine tune transcriptional regulation by these small viruses. Although it mainly confirms previously acquired data using standard methodology, it provides new insights to the applications of RNA-seq and by combining mRNA-seq with small RNA-seq gives rather complete information about the transcriptional regulation of the virus. The manuscript is well written, the data clearly presented and the conclusion drawn are adequate.However, there are some minor comments this reviewer would like to raise, which could clarify the complex data set and their interpretation:
Fig. 4: It would be helpful to add numbers to show the proportion of the individual transcripts for the individual proteins as well as the relative expression level of the 7 proteins into the figure.
Is there anything known about the stability/turnover of the individual proteins (e.g. non-structural proteins) and if so, could it be discussed in response to the transcriptional regulation?
Is there anything known about the properties of BocaSR (e.g. ORF, repeats, symmetries)?
How does the non-canonical poly(A) signal of (pA)dREH look like?
Author Response
Reviewer #1:
Critique 1: “Fig. 4: It would be helpful to add numbers to show the proportion of the individual transcripts for the individual proteins as well as the relative expression level of the 7 proteins into the figure. Is there anything known about the stability/turnover of the individual proteins (e.g. non-structural proteins) and if so, could it be discussed in response to the transcriptional regulation?”
Response: We added the percentage of the individual transcripts (%) and their length in the new Figure 4.
Due to lack of a single antibody that recognize all the NS and NP proteins, it is hard to determine the relative expression levels of the 7 viral nonstructural proteins. However, we had detected the expression of the 4 nonstructural proteins (NS1-4) in HBoV1-infected HAE cells using an in-house made anti-C-terminus antibody. We found that NS2/3 and NS4 were expressed at much a higher level than the NS1 protein, though NS1-coding mRNA is expressed at a level higher than those of the NS2-4. Nevertheless, we could not distinguish the expression levels between NS2 and NS3 as they have close molecular weights. We have discussed those points in lines 366-372, page 13 in the Discussion part.
Critique 2: “Is there anything known about the properties of BocaSR (e.g. ORF, repeats, symmetries)?”
Response: We had carefully studied the function of BocaSR in regulating HBoV1 replication (see reference 26). We found that BocaSR regulates the expression of HBoV1-encoded nonstructural proteins NS1, NS2, NS3, and NP1, but not NS4. BocaSR accumulates in the viral DNA replication centers within the nucleus and also plays a direct role in replication of the viral DNA. We have added this information in lines 374-376, page 13 in the Discussion part.
Critique 3: “How does the non-canonical poly(A) signal of (pA)dREH look like?”
Response: There is no degenerative AAUAAA hexamer presented in the (pA)dREH site. We believe other cis-signals, for example, upstream or downstream elements of polyadenylation plays an important role in defining the (pA)dREH site.
Reviewer 2 Report
Zou and co-workers present a straightforward work in this paper about the transcription profile including both mRNA and small RNA transcripts, of HBoV1 in HBoV1-infected HAE-ALI cultures. Viral transcripts were analyzed with both mRNA-seq and small RNA-seq methods. They identified novel transcription start and termination sites, and confirmed the previously identified splicing events. Importantly, an additional proximal polyadenylation site (pA)p2 and a new distal polyadenylation site (pA)dREH lying on the right-hand hairpin (REH) of the HBoV1 genome were identified in processing viral pre-mRNA.
The work is technically sound, the data gained is valuable, and it can serve as a future reference for further investigations of the molecular biology of HBoV1.
The problems I found with the work can easily be amended.
A general observation: the figures in the paper are presented in horrible quality, and that reflects (unnecessarily) badly on the work. So the graphics, the resolution and in general, the quality of all the figures must be improved.
From the mRNA-seq reads mapped to the HBoV1 full-length genome, it showed that viral mRNA transcripts initiated as early as nt 80, but at an abundance much lower than reads initiating at 291-296 (Figure 2B).
According to figure 2b, at 291-296 less than 50 reads is initiated, so there is no significant difference in the reads around nucleotides 80 and 295. Correct the text and/or the figure. I suggest to show coverage from nucleotide 50 to 350 (not 300) because the reads shoot up somewhere between nucleotide 300 and 400 (at least in the bad quality figure you present in 2a.) Please, put bars at every 500 nucleotides (not only at 1000) in Figure 2a.
Due to the fact that different HBoV1 mRNA transcripts share the same exon sequences, but the
237 alternative usage of the introns and alternative polyadenylation sites, it is hard to assemble de nove
238 HBoV1 mRNA transcripts, based on the mRNA-seq reads. Thus, we used the newly identified (pA)p2
239 and (pA)dREH and the previous identified mRNA transcripts as templates to assemble HBoV1 mRNA
240 transcripts using the mRNA-seq data.
It is a very important piece of information about the uncertainty of the results regarding ratio and even existence of the newly predicted mRNAs. This uncertainty should be further discussed in the discussion part.
I would like to see something like this in the discussion part: “Long-read RNA sequencing and/or the sequencing of (gel)separated mRNAs are needed to unambiguously prove the existence (and abundance) of the predicted mRNAs“ (of course in your own words).
Table 3 and Figure 4 are the core of this paper. They have to be as informative and digestible as possible. I do think that this paper can serve as a reference work for later studies, so as a professional courtesy, please, present your work on a more reader-friendly way.
Please, give us the calculated mRNA levels in percentage to each different mRNA (you gave only actual numbers) as you promise in the title of Table 3. Also, please, assign a name to each predicted mRNA (yes, 44 names): their predicted length (calculating from the most abundant start site to the different polyA sites ) number of exons with numbered exon borders. To help the reader, please make Figure 4 broader, and use guiding lines labelled with donor and acceptor site numbers. Please, also indicate and name the mRNAs whose existence were proven in earlier works including mRNAs for VP1, VP2, and VP3 (not only Vp).
Nevertheless, all these mRNA transcripts
320 processed at these novel splice sites are predicated not to produce novel proteins as they do not have
321 novel ORFs. We speculate that they may serve as backup mRNA transcripts that will be used in case
322 the original donor or acceptor sites are mutated during virus replication. Therefore, these alternative
323 donor and acceptor sites can still generate mRNA transcripts for production of viral proteins in the
324 events that the viral genome has mutations.
That is not so convincing. Please, consider the abundance of newly predicted mRNAs generated by the novel splice sites and compare their abundance to that of the already known mRNAs. The presence of different splice sites can be the consequence of simple fine tuning of the abundance of the mRNAs pool to ensure that different viral proteins are translated in optimal amounts during replication. On a related note: you did not investigate the time dependence of the splicing pattern, although in theory and in practice different splice sites can be used in different phases of the viral replication (I would check the literature). Different splice cites can be used in different cell types (HBoV1 may replicate in other tissues, even if it has not yet been proven). I think these are much more convincing arguments than the ones you gave here.
It has been reported that HBoV1 mRNAs, which were generated from transfection of a full345
length HBoV1 clone in HEK293 cells, were polyadenylated dominantly at the (pA)dREH site at nt 5,445
346 [36]. However, analyses of the mRNA-seq on the RNA samples derived from HBoV1 infection in
347 HAE-ALI suggests that the majority of HBoV1 mRNAs are distally polyadenylated at (pA)d1, with a
348 ratio of the total RNA reads at (pA)d1 vs. (pA)dREH of 2.09. We speculate that viral mRNA transcripts
349 generated from a transfected plasmid is polyadenylated differently from those produced during
350 virus infection.
It is possible, but what about the cell type differences?
HBoV1 small RNA transcripts. Recently, we identified a HBoV1 long noncoding RNA (BocaSR)
366 located between nt 5,199-5,338, which plays an important role in HBoV1 replication [26].
I guess you are talking about bocavirus-transcribed small RNA, and not about the long noncoding RNA.